# What a Tick Can Tell a Doctor: Using the Human-Biting Tick in the Clinical Management of Tick-Borne Disease

**DOI:** 10.3390/jcm12206522

**Published:** 2023-10-14

**Authors:** Stephen M. Rich, Eric L. Siegel, Guang Xu

**Affiliations:** Laboratory of Medical Zoology, Department of Microbiology, University of Massachusetts, Amherst, MA 01002, USA; esiegel@umass.edu (E.L.S.); gxu@umass.edu (G.X.)

**Keywords:** human-biting tick, pathogen testing, tick-borne disease

## Abstract

With expanding concern about ticks, there is a general sense of uncertainty about the diagnosis and treatment of tick-borne diseases. The diagnosis process is often based on clinical judgment in conjunction with laboratory testing and can be pathogen specific. Treatments may require disease-dependent approaches, and co-infections complicate or increase the severity of the clinical picture. Measuring exposure indices in the tick has become popular among providers and their patients, though this practice is not universally understood, and certain public health agencies have voiced concerns regarding interpretation and rigor of testing. As many providers subscribe to or recommend these services to aid in pretest risk and exposure assessments, this work sought to clarify the role of pathogen testing human-biting ticks as a complement to the diagnostic pipeline and raises points that must be addressed through future research and interdisciplinary conversation. Future work is needed to develop quality control oversight for tick testing laboratories. Studies on the integration of tick testing with human cases to see how these services affect health outcomes are also needed. Alongside these, improvements in the quality and availability of diagnostics are of critical importance.

## 1. Introduction

For most of human history, ticks were primarily of agricultural and veterinary concern. However, with the emergence of Lyme borreliosis in the late 1970s, ticks have become the predominant vector of human disease in the United States, accounting for more than 75% of domestically reported cases of vector-borne diseases [1,2]. Providers and their patients now face a large and growing list of pathogens associated with various forms of human tick-borne diseases, including human granulocytic anaplasmosis, babesiosis, Rocky Mountain spotted fever, alpha gal syndrome, and others [3].

Tick-borne disease incidence has risen dramatically over the past decade, and reporting efforts aimed at characterizing the true burden of these diseases remain plagued by underestimation [4]. This is attributable to many factors, including misdiagnosed and undiagnosed cases, differing standards in reporting/notifying requirements and case definitions which change over time, and variations in awareness and understanding of disease pathology and etiology [5,6,7,8]. The magnitude of the difference between estimated and reported cases has increased over time, with estimates of Lyme disease cases in the United States (76% of all tick-borne disease cases; 476,000 annual cases) exceeding ten times reported numbers, and this trend is more severe with less prevalent emerging tick-borne diseases, such as anaplasmosis and babesiosis [5,9,10]. Changing landscapes and climate are driving the range expansion of ticks and their pathogens, ensuring that these emergent challenges are only likely to worsen [11,12].

Lyme disease alone is estimated to cost the healthcare system more than $1 billion annually, excluding the unaccounted burden of misdiagnosed, undiagnosed, and chronic cases [13]. Confirmed and probable cases exceed an average of $1200 per patient with societal costs of around $2000 [14]. Increased costs are associated with more severe health outcomes in disseminated relative to localized disease, emphasizing the importance of accurate diagnosis in early disease stages [15]. The diagnosis of tick-borne diseases, however, remains notoriously difficult. Many common symptoms of Lyme disease are non-specific, such as joint and body aches, dizziness, and headaches [16]. The different pathogens that result in human disease also take many forms, such as those transmitted by the blacklegged tick, *Ixodes scapularis,* which include intraerythrocytic protozoa (*Babesia* spp.); granulocyte-targeting, obligate intracellular bacteria (*Anaplasma* spp.); flaviviruses (Powassan virus); and sparsely populating spirochetes (*Borrelia burgdorferi* and *B. miyamotoi*) [17,18,19,20]. Co-infections with these pathogens are common and complicate the clinical picture, as symptoms may be exacerbated, and multiplex diagnostic assays are not widely available [21]. The general availability and quality of diagnostic tests for these pathogens are also limited and often unstandardized, despite infections commonly associated with fatal outcomes in certain risk groups, and result in secondary disease of major systems (neuro, renal, and others) [22].

Due to limitations of the current state of tick-borne disease diagnostics, the clinical approach to diagnosis—including whether to test and the interpretation of test results—is centered around pretest probability, with a particular emphasis on exposure history in conjunction with clinical signs [23]. Tick-borne disease exposure is typically inferred from time spent in a disease-endemic region, either in travel or residence, with high-risk activities for labor or recreation [24]. Though immature stages of ticks are small and may go unnoticed throughout feeding, many patients present to a clinician aware of a tick bite, often after seeing and removing the feeding tick on their own [5,25]. Looking towards the human-biting tick for information on exposure has become increasingly popular among providers and their patients. Several public and private labs offer testing of human-biting ticks for pathogens, but the results and utility of these tests are not always understood in the context of the clinical picture. Nonetheless, tens of thousands of people subscribe to these services, often for a fee and following their provider’s recommendation. Aggregated human-biting tick data and observations provide insight into human-tick-pathogen encounters in passive surveillance systems [5,26,27,28,29]. But there remains an earnest need to clarify the role of the pathogen testing of human-biting ticks in human tick-borne disease risk assessment. 

## 2. The Current State of Tick-Borne Disease Diagnostics: Haves and Needs

Diagnosing tick-borne diseases is complicated, and available diagnostic tools to aid providers are still imperfect. The ‘correct’ approach to diagnosis is a polarizing topic, and recommendations vary with major advising organizations. Examples of these include the proceedings of the Infectious Diseases Society of America (IDSA), which provide very conservative guidelines, the International Lyme and Associated Diseases Society (ILADS), which challenge IDSA guidelines with approaches centered around the patient experience and complete background, and the National Institute for Health and Care Excellence (NICE), which primarily advise European activity with an evidence-based middle ground [22,30,31].

### 2.1. Lyme Disease Diagnostics

The most common approach for diagnosing Lyme disease follows a two-tiered serological pathway, which assesses the patient’s IgM and IgG antibody response to *B. burgdorferi* antigens [23,32]. This pathway can achieve a sensitivity of 70% and a specificity of 95% with an enzyme immunoassay (EIA) or immunofluorescence assay (IFA), followed by confirmatory Western blots targeting IgM and IgG if symptoms have appeared within 30 days or IgG-only if symptoms have persisted for longer than 30 days [33,34]. PCR and culture diagnostics are not standardized and are rarely used due to low sensitivity, contamination issues, and labor and technical intensity [34,35]. The process of diagnosis may also be affected by secondary manifestations of cardiac, rheumatologic, neurologic diseases and diseases of other systems which may present with Lyme disease [22].

The limited reliability of available diagnostics must be considered when performing and interpreting the serological-based tests for Lyme disease, particularly in early Lyme disease when diagnosis is most crucial to health outcomes. These are largely governed by antibody response, which is variable at the individual level [36]. Further, antibodies may take weeks to months to develop, so false negative results often show in cases of early infection despite the presence of symptoms [37]. Other considerations, such as background seropositivity resulting in false positive results due to antibody persistence, the potential for re-infection, and others, require attention when interpreting tests [38,39]. Because of these concerns, the testing decision and interpretation focuses heavily on pretest probability, with particular emphasis on exposure history considering activity and certain behaviors in endemic regions looked at alongside clinical signs [23].

Erythema migrans (EMs), the characteristic “bullseye” rash often observed during the early stages of Lyme disease (typically appearing within days to a month after a tick bite), are commonly used to guide testing and, in many cases, direct treatment without testing [40]. However, it is worth noting that 20–30% of patients bitten by ticks do not develop EMs, and there are various other conditions that can cause similar-looking skin lesions (such as southern tick-associated rash illness) [41,42]. Furthermore, treatment of EM-indicated Lyme disease with doxycycline may not cover common co-infections associated with other tick-borne diseases. This is the case with babesiosis caused by *B. microti*, which does not respond to doxycycline and requires atovaquone plus azithromycin (mild cases) or clindamycin plus quinine (severe cases) [43]. Treating Lyme disease based on EM presence therefore risks increased adverse health outcomes arising from unidentified co-infections. 

### 2.2. Other Tick-Borne Diseases

Lyme disease is the most prevalent tick-borne disease in the United States, but its vector *I. scapularis* transmits a number of other pathogens that can infect on their own or in co-morbidities with *B. burgdorferi* [44]. Anaplasmosis, babesiosis, and others are associated with symptoms which can overlap with Lyme disease [3]. In cases where patients are co-infected with *B. burgdorferi* and the etiological agents of these less prevalent diseases, symptoms of Lyme disease are often more severe, last much longer, and can be very difficult to diagnose [45]. Detection largely remains pathogen specific. *Babesia* spp., *Anaplasma phagocytophilum*, *B. miyamotoi*, and Powassan virus, are typically detected under light microscopy (*Babesia* and *Anaplasma*), using serological testing (all four), or nucleic acid amplification tests (NAATs, all four) [17,18,46].

Optimal detection assays for many of these pathogens have yet to be developed and therefore are largely unstandardized, leading to pathogen- and test-specific limitations regarding measures of accuracy and interpretation. Further, availability is inconsistent and may not be easily accessible for many providers. Multiplex assays which can assess the presence of many of these pathogens at once have been developed and implemented in some healthcare systems but are still not easily accessible or well developed [47]. Immune response throughout the course of infection with these pathogens also varies, and the optimal detection method across this timeline has yet to be determined for each of these pathogens [23].

## 3. Exposure Indices Drawn from the Human-Biting Tick and Their Relevance in Risk Assessment

From a clinical standpoint, the incubation period—from tick bite to disease development—is typically guessed at by providers relying on recollection of the patient. But if the tick is available, it can reduce speculation and provide an evidence base for pretest probability assessments. The tick’s anatomy is a record of exposure that can guide clinical decision making and encourage patient–provider communication before symptoms arise.

### 3.1. Integrating Species, Pathogen Detection, and Feeding Time into an Integrative Measure of Exposure

Infection and feeding in the tick serve together as proxies for risk and can be the earliest and most sensitive indicators of pathogen exposure. Additionally, species identification, typically performed under light microscopy or with molecular means, can be taken into consideration to determine the diseases that may be included in a differential diagnosis [48]. These should not be disregarded in the context of clinical diagnosis and may be considered of similar value to other circumstantial factors of exposure, such as geographic location (endemic regions) and time of year, which are recognized to hold value [3,23]. To be clear, consideration of these exposure measures is not to be misconstrued as a replacement for a diagnosis. Clinical signs of disease (such as EMs in the case of Lyme disease) and serological evidence do not manifest until weeks after the bite. As such, measures in the tick can be used as a vital piece of the clinical algorithm, guiding the vigilant patient and provider toward the risks of concern during the critical wait and watch period following tick removal (Figure 1).

The greatest value of tick pathogen testing comes with results pointing to a high-risk scenario consisting of three major indicators: (1) the tick was identified as belonging to a pathogen-harboring species; (2) the presence of one or more pathogens was shown in the tick; and (3) the tick has fed for a long enough duration to increase the risk of pathogen transmission. In this (common) situation, information can be used alongside other pertinent information, such as clinical signs and other known exposure information, to guide testing and interpretations. In cases where these indices do not point to high-risk measures, the patient and provider should not be complacent and appropriate monitoring for symptoms and open communication should persist. It is always possible that a second, unnoticed tick fed on the patient and therefore could still transmit disease.

### 3.2. Tick Testing and Feeding Physiology

In its simplest form, human-biting tick testing consists of nucleic acid extraction and amplification of genetic material with rapid and highly sensitive real-time polymerase chain reaction (PCR) assays [27]. The value in learning the tick’s infection status derives from the fact that while all tick-borne diseases start with a human-biting tick, not all human-biting ticks carry pathogens. Appreciation of the role of human-biting tick testing in assessing individual risk requires an understanding of the biological processes that occur from bite to pathogen transmission. To start, however, the broad scope of tick feeding physiology and tick-borne disease etiology must be distinguished from those other hematophagous arthropods. Mosquitoes, for example, feed for mere seconds after landing on a host and disperse, and they may bite several different hosts in a day [49]. Transmission of mosquito-borne zoonotic pathogens, present at low prevalence rates in mosquitoes (*Plasmodium* spp., West Nile virus, and others), therefore occurs within seconds [50]. Tick-human interactions are very different. Ticks in comparison must feed for several consecutive days on the same host and transmit pathogens which are present in much higher population prevalence [40,51].

### 3.3. Feeding Duration and Pathogen Transmission

Ticks are not merely tiny syringes that suck blood and spit saliva within a bite site. The feeding lesion formed is a complex microenvironment maintained from attachment until repletion [52]. Here, host and vector fluids combine to create a soup that will be slurped by the tick in the last period of feeding [53]. The appearance of pathogens, such as the Lyme disease spirochete, within the feeding lesion takes time. Once the feeding process begins, pathogens contained in the tick midgut begin traversing the epithelial wall to the open circulatory system, and finally penetrate the salivary glands for deposition into the feeding lesion [52]. Therefore, ticks must feed for an extended period (36–48 h) to allow for pathogen transfer, adding a dimension to exposure not available with other vectors, such as mosquitoes. Some less common pathogens may complete this process with shorter (Powassan virus, which may transmit within 15 min of feeding) or longer feeding durations (*Babesia* spp., which may take upwards of 72 h) [54]. This provides bite victims an opportunity to remove a feeding tick before it can transfer pathogens and present them for pathogen testing to aid in risk assessment. This is not possible or practical with mosquitoes which feed relatively quickly and have very low rates of infection.

Self-reported feeding times are well known to be drastically underestimated, particularly with immature tick (larval and nymphal) life stages [55]. In the laboratory, while pathogen status is inferred from PCR results, feeding times are estimated by physically measuring the tick’s body as a direct indication of cuticle distension. This is performed according to the scutal index (the ratio of the length of the idiosoma to the width of the scutum) and the coxal index (the ratio of the distance between the basal coxae of the fourth leg to the width of the scutum) [56]. Exposure risk changes with feeding time and visible changes in tick anatomy (Figure 2). The changing anatomy of the tick is therefore, in essence, a timeline of the infectious process. A combination of the two measures (coxal index, most useful in the first 24 h of feeding, and scutal in later stages) could be used to estimate the amount of time a tick has fed and therefore aid in risk assessment. As with species identification, providers could be trained to assess a tick’s degree of feeding and translate this into an exposure measure, but only a trained entomologist may be able to do so with accuracy. Future studies are needed to determine the feasibility of this physician-directed approach. 

## 4. The Controversy around Pathogen Testing of Human-Biting Ticks and the Path Forward

The Centers for Disease Control and Prevention (CDC) encourages the identification of tick species in good practice, as different species are associated with different pathogens, and these species often overlap in geographical range [57]. They do not, however, recommend pathogen testing of human-biting ticks [58]. This presents a seeming contradiction, as it conveys the utility of knowing some level of exposure information with the tick vector status at the population level but not at the individual tick level where pathogen presence varies. CDC also warns that pathogen testing lacks quality control standards, highlighting that no regulatory authority exists to ensure good practice [25]. This is of legitimate concern, however, only if one believes in the value of accurate results.

Even where correct determination of human tick infection is made, CDC also admonishes against misleading interpretation of positive and negative results. First, they warn that a pathogen positive human-biting tick does not necessarily indicate that that tick transmitted its pathogen load to the human host. To present this as a formal argument, the CDC holds that the bite of a pathogen-positive human tick may be a necessary but not sufficient cause by itself leading to infection. More specifically, the risk of infection is not necessarily a consequence of the presented threat of an infected tick. We agree entirely with this logic but using the common parlance of epidemiological models, we also know that risk is understood as a confluence of a present threat and some measure of the host exposure to that threat. In the case of tick-borne diseases, the presence of a pathogen-positive tick indicates that a threat is present, but additional factors of exposure bear on whether that threat conveys risk to the human subject. As explained in preceding paragraphs, this exposure can be measured unambiguously and independent of the threat assessment by evaluating the duration of the tick’s feeding and its corresponding changes in anatomy. Careful assessment of the threat and exposure can be incorporated into diagnostic algorithms as a contributing component.

This brings us to the CDC’s other major concern which is misinterpreting a negative result even when correct. Here, the apprehension is that the subject may be lured into a false sense of security by a negative tick test since they may have been bitten by an additional tick (or ticks) that was infected and went undetected. This is a weak and logically contradictory position since it implies that the flaw lies not in testing a tick, but in the shortcomings associated with failing to detect and test all ticks that a human subject might encounter. And so, while this disputation does not argue directly against the utility of testing human-biting ticks, its cautionary note is important. Failing to detect human-biting ticks is an important contributing factor to disease risk, and hence, consistent public health messaging on the importance of regular and thorough examination of people and pets for the presence of human-biting ticks is needed. We speculate that knowing the valuable information pertaining to risk contained within those ticks may reinforce the habit of scrutinizing for human-biting ticks. 

The strongest criticism of pathogen testing of human-biting ticks lies in the lack of transparent and rigorous standards of its practice. But rather than disregard tick testing based on the lack of quality control standards, a constructive approach for improvement would foster collaboration among the public health agencies, microbiologists, and entomologists to establish a certifiable, regulated framework for quality assurance. Regarding absent standards or regulatory oversight, the best assurance of quality is to subject human-biting tick testing methods and aggregated results to peer review. Many labs publish methods and collections of results, and even the staunchest critics of human-biting tick testing find value in this passive surveillance detailing human-tick-pathogen encounters, for which clinical data and entomological surveys of non-human-biting ticks cannot account [5,26,27,28,59]. These systems are well valued in early detection and endemic monitoring systems, detailing the spatiotemporal trends in novel, emerging, and endemic diseases.

## 5. Conclusions and Outlook

That tick-borne diseases pose serious risk to human health is not a matter of contention, but quantifying that risk is problematic. There are many avenues which require work to help mitigate the burden of tick-borne diseases on human health systems. Primarily, the quality and availability of diagnostics are far from what is needed. Together, these greatly weaken the correspondence between reported and estimated cases. Diagnosis of tick-borne diseases is often clinical and variable with respect to pathogens. As co-infections are becoming more common and complicate the clinical picture, more cases may be misdiagnosed.

When performed with the requisite quality control systems and appropriate interpretation of results, the promise of human-biting tick testing can be the early assessment of exposure that precedes clinical disease. The practice of looking towards pathogen-harboring vectors or reservoirs is not something new. The handling of the rabies virus is a very prominent example. Clinical signs appear when pathogens reach the human brain [60]. Disease at this point is almost always fatal, and treatment is limited to supportive care [61]. Skunks, raccoons, foxes, bats, and high-risk companion animals that bite humans are often subject to prompt euthanasia and post-mortem testing on brain tissue [62]. The rabies virus presence in this test is often used as the foundation for guiding post-exposure prophylaxis. Here, the rabies virus test is analogous to the pathogen test in the tick, informing on exposure which precedes disease.

Significant improvements are needed in many areas of tick-borne disease diagnostics, especially regarding precision, accessibility, and reporting consistency. This is becoming particularly important with emerging and rare pathogens, such as Powassan Virus. With this, the emergence of -omics, personalized medicine, and other progressive approaches to clinical medicine hold promise in the future of diagnosis and treatment [63,64]. Human-biting tick testing can provide support considering these issues and foster engagement opportunities between providers and patients, and conversations may extend beyond pathogen results to practices of personal protection as the basis of disease prevention and other educational topics. Careful determination of risk, determined by assessing threat and exposure (see Section 4), can be incorporated into the diagnostic algorithm (Figure 1). Model algorithms such as this one can and should be rigorously tested empirically to determine whether following them can favorably improve patient outcomes following encounters with human-biting ticks. Ignoring critical components of risk is not sound clinical practice.

Recommendations against the utility of human-biting tick testing, often couched in contradiction, are oversimplistic and may lead to a missed opportunity to improve a promising tool in the effort to reverse the trend in tick-borne diseases. Future work is needed to establish quality control oversight and build awareness around the proper use of these programs. Further studies should also seek to evaluate whether human-biting tick testing can improve disease outcomes with studies involving follow-up in those that utilize tick testing services. In light of the incredible economic burden posed by tick-borne diseases, it is beneficial to assess approaches which may mitigate some of this burden. Pathogen testing is inexpensive relative to the costs of chronic care and acute, progressive disease and may hold promise in promoting favorable health outcomes. Subsidization and standardization of tick testing assays and surveillance systems should become a priority, however, the limited funding available for direction towards ticks and tick-borne diseases relative to mosquito-borne and other diseases poses a significant challenge.

## Figures and Tables

**Figure 1 jcm-12-06522-f001:**
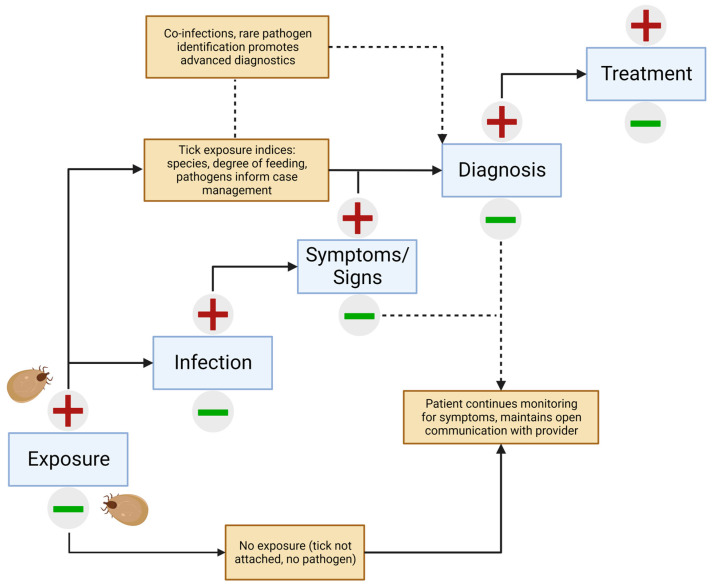
Algorithm integrating pathogen testing of human-biting ticks with the clinical presentation of tick-borne disease.

**Figure 2 jcm-12-06522-f002:**
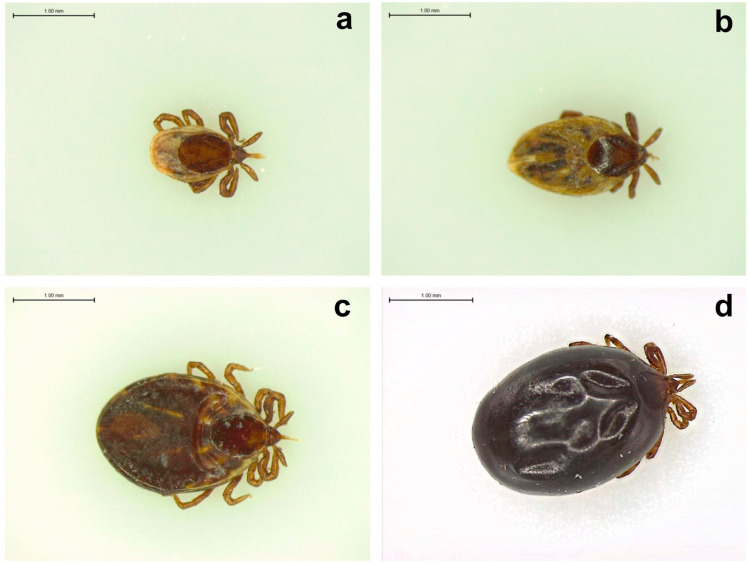
Human-biting *Ixodes scapularis* nymphal ticks removed at different stages of feeding as viewed with light microscopy: (**a**) unfed/flat; (**b**,**c**) partially fed; (**d**) engorged/replete. The scale bar in each image represents 1.00 mm length.

## Data Availability

Not applicable.

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
