# Peer review of "What a Tick Can Tell a Doctor: Using the Human-Biting Tick in the Clinical Management of Tick-Borne Disease"

_jcm, 2023, doi:10.3390/jcm12206522_

Round 1

Reviewer 1 Report

If according to lines 208-209 (Self-reported feeding times are well-known to be drastically underestimated, particularly with immature tick (larval and nymphal life stages), how much the „combination of the two measures (coxal index, most useful in the first 24 hours of feeding; and scutal in later stages) could be used to estimate the amount of time a tick has fed and therefore aid in risk assessment”? 

This might be a roadblock considering the usefulness of indexes applied only in the adult ticks. The coxal and scutal indexes may be used in a diagnosis score but the probability of great improvement would be, presumably modest.

On the other hand, I agree with the authors that the „Recommendations against the utility of human-biting tick testing, often couched in contradiction, are over simplistic and may lead to a missed opportunity to improve a promising tool in the effort to reverse the trend in tick-borne disease”.

Line 82 ILADS (Society not ociety)

Author Response

We acknowledge the reviewers concerns about the limitations of the dual index approach.  Clearly a trained entomological tech will be able to do better than the average tick bite victim.  

More importantly, we are grateful for the reviewers acknowledgement of the validity of our argument with respect to the contradictory recommendations. 

Reviewer 2 Report

It is an excellent viewpoint article considering the threat to public health and heavy economic burden posed by tick-borne diseases. As an emerging technic, metagenomic next-generation sequencing (mNGS) has been used to detect pathogens in ticks and patients with tick-borne diseases (PMID 28947798, 36000911, 36997767, and so forth.). Presenting and discussing the advantages and disadvantages of mNGS in diagnosing tick-borne diseases will strengthen this article. 

Author Response

We are grateful for the reviewers praise of the article. 

While we agree that mNGS approaches are likely to contribute much to our understanding, the present viewpoint is focused primarily on the rationale for tick testing, and is more agnostic to the particular modes of pathogen detection.  We have not, for example, evaluated the relative strength of conventional vs. real-time PCR.   We hope that this paper will stimulate dialogue that has been hampered by voices that dismiss the utility of testing human biting ticks out of hand. 

Reviewer 3 Report

Rich et al. present a paper supporting the idea of using results from testing human-biting ticks submitted by patients in clinical evaluations for tick-borne diseases.  They propose that information about the tick species identity, infection status and engorgement that may be collected with human-biting tick submissions may be helpful in guiding clinicians' decisions regarding diagnosis and decisions to treat or not treat patients for possible tick-borne disease infections.  They argue that the use of such information may be viewed similarly to that obtained from post-mortem testing of human-biting animals for the rabies virus in making clinical decisions on  post-exposure rabies prophylaxis.

The authors provide a good description of the challenges involved with Lyme disease diagnosis and tick-borne disease diagnostics.

The reasonings for an integrated species, pathogen, feeding time integrative measure of exposure (and in extension, risk for infection), I believe, are sound.  It makes sense to include the added information that may be provided from a human-biting tick in clinical decision making; just as simply hearing from a patient that they were recently bitten by a tick often is.

I believe the paper could be improved by further discussing the controversy around pathogen testing of human-biting ticks.  The authors include a short section on this on page 6 of 9 of the submitted paper.  They reference the CDC's objection to pathogen testing human-biting ticks, but don't provide the reasons for CDC's objection.  They discuss reasons for objection by Massachusetts Department of Health as associated with the lack of quality control standards.  I think this is also true for the CDC, but not only with regards to standards as might relate to laboratory processing of ticks, but also on how to standardize the interpretation of testing results.  As the authors describe, other factors are involved in the transmission of pathogens, including duration of tick feeding and the possibility that the tick collected is not the tick that may have transmitted an infection.  I think the authors just have to draw the connections between the controversy section and other sections of the paper and restate in this paragraph the need for an integrative index for using biting tick testing in clinical decision making.

It is also worth mentioning that in order for such methods to be widely adopted, methods would have to be adaptable to the clinical laboratory. Most clinicians will not be the ones to examine ticks under a microscope to speciate or to assess level of engorgement.  Rather, laboratory technicians would be involved in this.  Furthermore, most clinical labs will not be set up to do manual digestions of tick tissues for pre-DNA extraction methods.  To really be useful, authors have to be forward-thinking and consider ways that tick processing and testing can most easily be adopted by clinical laboratories.

Author Response

As recommended by the reviewer, we have expanded the discussion of controversy surrounding CDC admonition against tick testing. We thank the reveiwer for this recommendation whihc has strengthened the argument.